# Morusin Enhances Temozolomide Efficiency in GBM by Inducing Cytoplasmic Vacuolization and Endoplasmic Reticulum Stress

**DOI:** 10.3390/jcm11133662

**Published:** 2022-06-24

**Authors:** Rongchuan Zhao, Yuanshuai Zhou, Hong Zhang, Jinlin Pan, Fan Yang, Ruobing Zhang, Nafees Ahmad, Jiao Yang, Minxuan Sun

**Affiliations:** 1Suzhou Institute of Biomedical Engineering and Technology, Chinese Academy of Sciences, Suzhou 215163, China; zhaorc@mail.ustc.edu.cn (R.Z.); panjl@sibet.ac.cn (J.P.); zhangrb@sibet.ac.cn (R.Z.); 2School of Biomedical Engineering (Suzhou), Division of Life Sciences and Medicine, University of Science and Technology of China, Hefei 230026, China; zhouys@sibet.ac.cn; 3School of Life Sciences, Shanghai University, Shanghai 200444, China; 15856927048@163.com (H.Z.); yf1320999750@163.com (F.Y.); 4Institute of Biomedical and Genetic Engineering, Islamabad 44000, Pakistan; nafravian1@yahoo.com; 5Institute of Clinical Medicine Research, Suzhou Science & Technology Town Hospital, Gusu School, Nanjing Medical University, Suzhou 215163, China

**Keywords:** glioblastoma multiforme, morusin, cytoplasmic vacuolation, endoplasmic reticulum stress, tumor progression

## Abstract

Glioblastoma multiforme (GBM) is an aggressive brain tumor with high risks of recurrence and mortality. Chemoradiotherapy resistance has been considered a major factor contributing to the extremely poor prognosis of GBM patients. Therefore, there is an urgent need to develop highly effective therapeutic agents. Here, we demonstrate the anti-tumor effect of morusin, a typical prenylated flavonoid, in GBM through in vivo and in vitro models. Morusin showed selective cytotoxicity toward GBM cell lines without harming normal human astrocytes when the concentration was less than 20 µM. Morusin treatment significantly induced apoptosis of GBM cells, accompanied by the activation of endoplasmic reticulum (ER) stress, and the appearance of cytoplasmic vacuolation and autophagosomes in cells. Then, we found the ER stress activation and cytotoxicity of morusin were rescued by ER stress inhibitor 4-PBA. Furthermore, morusin arrested cell cycle at the G1 phase and inhibited cell proliferation of GBM cells through the Akt–mTOR–p70S6K pathway. Dysregulation of ERs and cell cycle in morusin exposed GBM cells were confirmed by RNA-seq analysis. Finally, we demonstrated the combination of morusin and TMZ remarkably enhanced ER stress and displayed a synergistic effect in GBM cells, and suppressed tumor progression in an orthotopic xenograft model. In conclusion, these findings reveal the toxicity of morusin to GBM cells and its ability to enhance drug sensitivity to TMZ, suggesting the potential application value of morusin in the development of therapeutic strategies for human GBM.

## 1. Introduction

Glioblastoma multiforme (GBM) is the most common and aggressive brain tumor in humans, which is clinically difficult to treat [1]. Temozolomide (TMZ) chemotherapy, in combination with surgery and radiotherapy, is the current standard care for GBM patients. Despite this multimodal approach, GBM inevitably relapses, and the median survival of GBM patients remains only about 15 months [2]. TMZ resistance severely limits efficacy and has become a significant cause of poor prognosis [3]. Therefore, there is an urgent need to identify novel therapies for GBM.

The endoplasmic reticulum (ER) participates in multiple cellular processes, including protein folding, Ca2+ storage, and lipid and carbohydrate metabolism [4]. In response to diverse cellular stresses, unfolded or misfolded proteins accumulate in the ER lumen, leading to ER stress [5,6,7]. To restore homeostasis, cells have an integrated system that includes the unfolded protein response (UPR), ER-associated degradation (ERAD), autophagy, hypoxic signaling and mitochondrial biogenesis. UPR plays a critical role in reversing ER stress and is regulated by three ER membrane-embedded sensors: double-stranded RNA-activated protein kinase (PKR)-like ER kinase (PERK) [8], activating transcription factor 6 (ATF6) and inositol-requiring enzyme 1 (IRE1) [9,10]. However, UPR activity must be tightly regulated, as a high level of or prolonged UPR signaling is associated with cell death. In contrast, more moderate UPR signaling enables cell homeostasis. Although direct involvement of the UPR in gliomagenesis has not been demonstrated yet, several pieces of evidence have indicated an important role for the UPR in GBM growth and progression [11,12,13].

Natural products are the most critical lead compounds which have been widely used to manufacture anti-cancer drugs [14]. Flavonoids are a significant part of natural products, including flavone, flavanone, isoflavone and dihydrochalcone. Morusin is a flavonoid derived from the root bark of Morus alba L. Morusin exhibits various biological activities, including antioxidant, antimicrobial and anti-inflammatory effects [15]. Previous studies also showed that morusin exhibits anti-tumor effects in various cancer types, including colorectal cancer, pancreatic cancer, gastric cancer, breast cancer, prostate cancer, cervical cancer, hepatocarcinoma and glioblastoma [16,17,18,19,20,21,22,23,24,25,26,27,28]. Several natural products, such as delta (9)-tetrahydrocannabinol (THC), piperlongumine and perillyl alcohol (POH) were reported to induce ER stress/UPR and cause cytotoxicity in GBM models. Whether morusin could activate ER stress in GBM is still not clear.

In this study, we demonstrated the selective killing effect of morusin on GBM both in vitro and in vivo. The GBM cell death showed strong correlations with strikingly enhanced ER stress and UPR induced by morusin. Additionally, we showed that morusin arrests the cell cycle at the G1/S phase and inhibits cell proliferation by suppressing the Akt–mTOR pathway. Moreover, morusin and TMZ have synergetic cytotoxicity against GBM both in vitro and in vivo, which may be a potential strategy for GBM therapy.

## 2. Materials and Methods

### 2.1. Cell Lines and Cell Culture

Human astrocytes, U87 and U251 cell lines, were obtained from the Chinese Academy of Sciences Cell Bank (Shanghai, China). Cells were cultured in high-glucose Dulbecco’s modified eagle’s medium (DMEM) (Hyclone, Logan, UT, USA) supplied with 10% fetal bovine serum (FBS) (Gibco, Waltham, MA, USA). The patient-derived primary GSCs, GSC111 and GSC123, were a kind gift from Peng Fu (The Tongji Hospital of Huazhong University of Science and Technology). GSC111 and GSC123 were cultured in DMEM F-12 (SIGMA, lot number RNBG2219) supplemented with EGF (20 ng/mL, Gibco, lot number PHG0311), bFGF (20 ng/mL, Gibco, lot number PHG0368), B27 (1×, Gibco, lot number 17504044) and NEAA (1×, Gibco, lot number 11140050). All cells were cultured at 37 °C with 5% CO_2_ and 95% humidity in a constant temperature incubator. Morusin (CFN97083) was purchased from ChemFaces (Wuhan, China) and dissolved in DMSO.

### 2.2. Cell Viability Assay

Cell viability was detected using a cell-titer blue kit (Promega, Madison, WI, USA). GBM cells (1 × 10^4^ cells/well) were seeded in 96-well plates and cultured for 24 h before treatment. Cells were treated with different concentrations of morusin, 4-PBA or TMZ for 48 h. CellTiter-Blue^®^ Reagent (20 µL/well) was added to plates and incubated for 2 h, and the fluorescence at 560/590 nm was recorded by a microplate reader (Thermo, VARIOSKAN LUX).

### 2.3. Real-Time Quantitative PCR

The total RNA of samples was extracted using an EZ-press RNA Purification Kit (EZBioscience, Roseville, MN, USA). A 4× Reverse Transcription Master Mix (EZBioscience, Roseville, MN, USA) was used for reverse transcription reaction at 42 °C for 15 min and 95 °C for 30 S. The 2× SYBR Green qPCR Master Mix (EZBioscience, Roseville, MN, USA) was used to perform qPCR following this protocol: denaturation (5 min, 95 °C) and 40 amplification cycles (10 s at 95 °C, and 30 s at 60 °C) by using a 7500 Fast Real-Time PCR System (Applied Biosystems, Bedford, MA, USA). GAPDH served as an internal control. The relative quantity of gene expression was analyzed by the 2−ΔΔCt method. The primers used for qPCR analyses are listed in Table 1. The qPCR had at least 3 biological replicates.

### 2.4. Western Blot

Cell lysates were prepared in RIPA lysis buffer, including protease inhibitors (CoWin Biosciences, Beijing, China) and phosphatase inhibitors (CoWin Biosciences, Beijing, China). Equal amounts of proteins were subjected to SDS-polyacrylamide gels. After that, the proteins were transferred to nitrocellulose membranes, and the membranes were blocked with 5% non-fat dried milk for 1 h at room temperature. Each membrane was then incubated with the primary antibody, anti-tubulin (1:1000, AbClonal, AC021), anti-cyclin D1 (1:1000, CST, CST2978), anti-PCNA (1:1000, CST, CST13110), anti-p-eIF2a (1:1000, Abclonal, AP0692), anti-Chop (1:1000, Abclonal, A0221), anti-ATF6 (1:1000, Abclonal, A0202), anti-AKT (1:1000, CST, CST4060), anti-p-AKT (1:1000, CST, CST4691), anti-mTOR (1:1000, Abclonal, A2445), anti-p-mTOR (1:1000, Abclonal, AP0115), anti-P70S6K (1:1000, Abclonal, A2190) or anti-p-P70S6K (1:1000, Abclonal, AP0478), overnight at 4 °C. After incubation with IRDye 800cw or 680cw conjugated antibodies (1:5000 dilution) for 1 h, the membranes were imaged with Odyssey^®^ CLx Infrared Imaging System. The WB had at least 3 biological replicates.

### 2.5. Scanning Electron Microscope (SEM)

Cells were fixed by 4% glutaraldehyde and post-fixed in 1% OsO4 in 0.1 M cacodylate buffer for 2 h. After being stained with 1% Millipore-filtered uranyl acetate, the samples were then dehydrated in increasing ethanol concentrations, and then infiltrated and embedded in epoxy resin (ZXBR, Spon 812). Electron photomicrographs of GBM cell ultra-structures were taken with a scanning electron microscope (JEM-1200EX II, JEOL; Tokyo, Japan).

### 2.6. Cell Cycle Assay

Cells were seeded in 10 cm dishes at 1 × 106 cells, and serum-starved using 1% FBS for 24 h for cycle synchronization before treatment with 0, 10 and 20 μM morusin for 24 h. Cells were harvested, fixed in 70% ethanol at 4 °C overnight and then incubated with DNA-interacting dye propidium iodide (PI) for 30 min at 37 °C. Cell cycle analysis was performed using a flow cytometer (BD FACSCelesta).

### 2.7. RNA-Sequencing Analysis

The total RNA of morusin (15 μM, 24 h)-treated U87 cells and normal control (NC) U87 cells without treatment was isolated and underwent quality control. The preparation of whole transcriptome libraries and deep sequencing were performed by Novogene Bioinformatics Technology Cooperation (Beijing, China). Raw reads were mapped to the homo sapiens’ reference genomes with HISAT2 software [29]. FeactureCounts was employed for quantification of mapped reads into genomic features [30]. The differential expression analysis was performed by limma package [31]. To assess the functional features of differentially expressed genes, clusterProfiler and GSEABase were applied for functional annotation [32] and enrichment analysis [33], respectively.

### 2.8. Generation of Orthotopic Xenograft Models

Male BALB/c nude mice of 8–10-week-old were purchased from SiPeiFu (Beijing) Biotechnology Co., Ltd., and housed under standard conditions at the animal care facility at the Center of Experimental Animals of Suzhou Institute of Biomedical Engineering and Technology, Chinese Academy of Sciences. Briefly, using a stereotactic frame, 1 × 10^6^ U87 cells were stereotactically injected into the right putamen (1 mm forward, 2 mm right lateral from the bregma and 3 mm down from the dura). Morusin (20 mg/kg) and TMZ (40 mg/kg) were subcutaneously injected 2 weeks after U87 inoculation. The control group were injected with 0.1% DMSO diluted in PBS. The body weight of each mouse was measured every three days. When related neuropathological signs developed, mice were sacrificed and perfused with PBS and 4% paraformaldehyde (PFA). Mice brains were dissected and fixed in 4% PFA for 24 h. The procedure has been approved by Animal Care and Use Committee, Suzhou Institute of Biomedical Engineering and Technology, Chinese Academy of Sciences.

### 2.9. Immunohistochemical (IHC) Staining

For paraffin slides from the tissue of patients, after deparaffinization and hydration, slides were repaired by boiling in Tris-EDTA buffer (PH = 8.0) for 10 min and then washed with PBS. Frozen tissue slides of xenograft were washed with PBS three times. Next, sections were treated with 3% H_2_O_2_ for 20 min to bleach endogenous peroxidase. After blocking with donkey serum in PBS for 30 min, slides were incubated overnight at 4 °C overnight with primary antibodies. Following washes with PBS, slides were then incubated with HRP-conjugated donkey anti-rabbit or mouse secondary antibody (DAKO) for 45 min at room temperature. Sections were then stained by DAB with Gill hematoxylin counterstaining. Samples incubated without primary antibodies were used as negative controls.

### 2.10. Statistical Analysis

All data are presented as means ± s.e.m. Statistical differences between two groups were evaluated using a 2-tailed *t*-test. Overall survival (OS) was plotted by the Kaplan–Meier method and compared by the log-rank test. Statistical analysis was performed using GraphPad Prism 8.0. Statistical significance was defined as: * *p* < 0.05, ** *p* < 0.01, *** *p* < 0.001.

## 3. Results

### 3.1. Selective Cytotoxicity of Morusin toward GBM Cells and GSC Cells

Morusin, one isoprene flavonoid derivative in the root bark of Morus, has the chemical structure of a flavone with hydroxyl groups at C-5, C-2′ and C-4′, as shown in Figure 1A. To detect the cytotoxic effect of morusin on GBM cells, a GBM cell line and GBM stem cell lines were treated with different doses of morusin (0, 5, 10, 20, 40 μM) for 24 h, and cell viabilities were detected. Morusin had almost no effect on the viability of normal human astrocytes at low concentrations (0, 5, 10, 20 μM), but significantly suppressed the viabilities of U87, U251, GSC111 and GSC123 cells in a dose-dependent manner (Figure 1B). When the concentration of morusin is as high as 40 μM, it shows strong toxicity to normal human astrocyte and reduced cell viability by more than 50% (Figure 1B).

The cytotoxicity of morusin to U87 and U251 cells was also confirmed by live/dead assay and annexin-V/PI staining assay. After morusin treatment for 24 h, the proportions of dead cells among both U87 and U251 cells were strikingly elevated (Figure 1C,D). The proportions of apoptotic cells (annexin V+) and dead cells (PI+) were increased by morusin, and the percentage of double negative live cells was decreased (Figure 1E,F). Consistently, the number of viable cells was also significantly decreased 48 h after morusin removal (Figure 1G), indicating that morusin has a long-term inhibitory effect on GBM cell proliferation. The above results suggest that morusin had selective toxicity to GBM cells. It decreased cell viability, inhibited cell proliferation and induced cell apoptosis of GBM cells.

### 3.2. Morusin-Induced Cytoplasmic Vacuolation and Activated ER Stress in Human GBM Cells

After morusin treatment, high densities of vesicles were found in U251 and U87 cells under a phase contrast microscope (Figure 1H,I). In order to observe the intracellular structure more clearly, morusin-treated GBM cells were further visualized using electron microscopy. We found morusin-induced cytoplasmic vacuolation in U251 and U87 cells, which was rarely seen in the control group (Figure 2A). To investigate whether morusin induced cytoplasmic vacuolation in GBM cells through ER dilation, we stained the ER of both the untreated and 20 μM morusin-treated U87 cells with the ER tracker dye. We found that vesicles’ membranes appeared positive for the ER-specific marker, suggesting that morusin-induced cytoplasmic vesicles originated from ER (Figure 2B). Given that cell death associated with cytoplasmic vacuolation is usually associated with persistent ER stress [7], we performed qPCR and Western blot to evaluate the activation of ER stress pathway associated molecules. Morusin remarkably induced the expression of phosphorylated eukaryotic initiation factor (p-eIF2α), activating transcription factor 6 (ATF6) and C/EBP homologous protein (CHOP) in a dose-dependent manner, in both U87 and U251 cells (Figure 2C–F). Likewise, immunofluorescence staining of ER stress marker CHOP in U87 also indicated the elevated ER stress and UPR by morusin (Figure 2G). UPR is tightly linked to autophagy, a major cellular catabolic process that sequesters large protein aggregates and damaged organelles for degradation in autophagosomes [34]. Our results also confirmed the existence of autophagosomes in GBM cells after morusin treatment (Figure 2H), which is strong evidence for the activation of autophagy. Taken together, these results indicate that ER stress activation and autophagy were involved in GBM cells death and cytoplasmic vacuolation caused by morusin.

### 3.3. 4-PBA Rescued the Cytotoxicity of Morusin in GBM Cells

To validate whether morusin regulates the death of GBM cells through ER stress, 4-PBA was employed to inhibit the activation of ER stress. As expected, morusin-induced cytoplasmic vesicles in two GBM cell lines were markedly decreased (Figure 3A), which indicated that ER stress and UPR in the presence of morusin were inhibited effectively by 4-PBA. Consistent with morphology recovery, the cell viability of U87 and U251 cells suppressed by morusin was partially rescued by 4-PBA (Figure 3B,C). What is more, 4-PBA reversed the up-regulation of p-elF2a and CHOP by morusin in U87 and U251 (Figure 3D,E). Briefly, these results demonstrate that morusin-induced cell death in GBM cells is partially attributable to ER stress and UPR.

### 3.4. Morusin Caused Cell Cycle Arrest and Inhibited Cell Proliferation through Akt–mTOR–p70S6K Pathway

The above findings preliminarily clarified that the dysregulation of ERs and UPR by morusin contributes to the killing effect on human GBM cell lines. To further explore the anti-cancer mechanism of morusin beyond ERs and UPR, flow cytometry was performed to analyze the alteration of the cell cycle. In two GBM cell lines, morusin caused cell cycle arrest at G1 (increased cell ratios in G1 and decreased cells ratios in S) (Figure 4A,B). A stable cell cycle is necessary for cell growth and proliferation. Therefore, the expression levels of proteins related to cell proliferation were detected by Western blotting. The expression of cyclin D1 and PCNA were acutely inhibited by morusin (Figure 4C,D). Ki67+ U87 cells were also markedly decreased compared with the control group (Figure 4E,F). These results demonstrate that morusin treatment perturbs cell cycle arrest at the G1 phase and inhibits the expression of cell proliferation-related proteins.

Subsequently, to further understand the mechanism by which morusin inhibits cell proliferation, Western blotting was performed to detect the alteration of the Akt–mTOR pathway in U87 and U251 cells. Our data showed that morusin treatment suppressed the expression of the phosphorylation of Akt, accompanied by decreased phosphorylation of mTOR and p70S6K (Figure 4G–J). Down-regulation of the ratio among p-Akt/Akt, p-mTOR/mTOR and p-P70S6K/P70S6K indicates the inhibition by morusin of cell proliferation. These results indicate that morusin not only induces ER stress and UPR, but also mediates anti-proliferative effects by inhibiting the Akt–mTOR pathway.

### 3.5. RNA-seq Analysis Revealed the Dysregulation of ERs and Cell Cycle Pathways in GBM Cells after Morusin Treatment

To validate in vitro findings, RNA sequencing was performed to identify differentially expressed genes induced by morusin. Compared with control cells, a total of 1516 up-regulated genes and 1887 down-regulated genes were identified in cells treated with morusin (|Fold change|>2, FDR< 0.05). Gene Ontology (GO) enrichment analysis showed that the up-regulated genes were enriched in response to ER stress, unfolded protein and other biological processes related to protein transport (Figure 5A). KEGG pathway enrichment analysis showed the up-regulated genes were associated with cytokine−cytokine receptor interactions and protein processing in the endoplasmic reticulum (Figure 5B). In addition, both GO and KEGG pathway enrichment analysis showed that the down-regulated genes were mainly involved in DNA replication and the cell cycle (Figure 5C,D). More importantly, Gene Sets Enrichment Analysis (GSEA) showed that the up-regulated genes in morusin-treated cells were enriched in signatures of protein export; the down-regulated genes were mainly involved in cell cycle regulation (Figure 5E,F). Finally, we validated the gene expression data on ERs and cell cycle (G1/S), and found that most ERs genes and cell cycle genes were up-regulated or down-regulated after morusin treatment, respectively (Figure 5G). Collectively, bioinformatics analysis confirmed in vitro results about the induced ER stress level and cell cycle arrest by morusin in GBM cell lines.

### 3.6. Combination of Morusin and Temozolomide Remarkably Induced ER Stress and Exhibited an Anti-GBM Effect Both In Vitro and In Vivo

Temozolomide (TMZ) is a widely used first-line treatment for GBM. To investigate whether morusin could enhance the anti-tumor activity of TMZ, we compared the effects of morusin and TMZ alone and in combination in vitro and in vivo. Cell viability assay showed that the combinations of different doses of TMZ with 20 μM morusin had pronounced killing effects on U87 (Figure 6A) and U251 (Figure 6B) cell lines. According to CI analysis, high doses of TMZ (100 μM) and morusin (20 μM) were synergistic in inhibiting cell viability of U87 (Figure 6C) and U251 (Figure 6D). To better understand the potential mechanism of the synergistic effect, we examined the ERs level after morusin and TMZ treatment. As expected, combinational treatment of morusin and TMZ dramatically induced the expression of p-eIF2a and CHOP (Figure 6E,F), which seemed to be a key factor in the synergistic cytotoxicity against GBM cells.

To further investigate the in vivo anti-tumor effects of morusin, we established a xenograft model of GBM through orthotopic inoculation of U87 cells into the brains of nude mice. The xenografted mice were randomly divided into four groups. They were subcutaneously injected with vehicle control, morusin (20 mg/kg), TMZ (40 mg/mL) and morusin + TMZ (morusin 20 mg/mL, TMZ 40 mg/mL) two weeks after U87 inoculation (every twice days, 7 times) (Figure 6G). Survival analysis indicated that morusin prolonged the overall survival of xenografted mice compared with the control group (*p*-Value = 0.0164) (Figure 6H). The combinational injection of TMZ and morusin improved the survival of mice compared with TMZ alone (*p*-value = 0.056) and vehicle control (*p*-Value = 0.0041). Hematoxylin and eosin (H&E) staining of brain slices revealed an obvious reduction in tumor mass in mice treated with morusin and TMZ compared with other groups (Figure 6I). Consistently with morusin and TMZ inhibiting GBM cell viability and inducing ER stress in vitro, we also demonstrated that combinational treatment with morusin and TMZ could trigger ER stress and suppress GBM progression in the xenograft model by CHOP and Ki67 staining, respectively (Figure 6J).

In summary, we demonstrated the anti-tumor effect of morusin by inducing cell death through ER stress and inhibiting cell proliferation through the AKT–mTOR–p70S6K pathway (Figure 7). Additionally, we confirmed that morusin could enhance temozolomide efficiency in GBM both in vitro and in vivo.

## 4. Discussion

Flavonoids have been proposed as potential chemotherapeutic agents against cancers. There are many phase II and phase III clinical trials using flavonoids as single agents, such as flavopiridol, flavone acetic acid (FAA), phenoxodiol, isoflavone and polyphenon E; or combined with other therapeutics against hematopoietic/lymphoid or solid cancer. Among the total 615 patients in 10 trails with hematopoietic or lymphoid tissue cancer treated with flavonoids, 140 patients achieved a complete response (CR), and 88 patients achieved a partial response (PR) [35]. One-hundred and thirty-nine patients with a CR were from clinical trials (350 patients) that used flavopiridol schema combined with a first-line drug schema therapy for treatment of acute myeloid leukemia [35]. In clinical trials for chronic lymphocytic leukemia, 29 out of 42 patients achieved PR after treatment of polyphenon E as a single agent [36]. There were also 53 chronic lymphocytic leukemia patients who achieved PR out of 183 patients for the treatment of flavopiridol as a single agent [35]. There were also several clinical trials of flavonoids in the treatment of solid cancer. For example, FAA was used for melanoma [37] and colon, breast and head and neck carcinoma patients [38]. Phenoxodiol was employed in patients with ovarian cancer [39], ovarian and peritoneal cancer [40]. Isoflavone was used for pancreatic and prostate cancer. Although the effects of flavonoids against solid cancer are not as good as those of flavonoids against hematopoietic/lymphoid cancer, combinations of flavonoids with first-line drugs may be potential therapy for cancer patients. However, a better understanding of flavonoid action is still required to achieve better outcomes in clinical tumor treatments.

The microenviromental conditions in which cancer cells exist are usually severe, such as hypoxia, hypoglycemia and low pH. These conditions often induce ER stress and affect intracellular protein production [6]. For example, hypoxia affects the disulfide forming process and leads to aberrant protein folding. Low glucose will reduce ATP production that is required for the protein folding machinery [41]. Therefore, UPR was considered to function primarily as an adaptive system to support the survival of tumor cells. Several studies have reported the increased endoplasmic reticulum stress and UPR activity in GBM. Compared with normal brain tissue, the levels of ER stress in different in vitro cultured GBM cell lines were up-regulated, such as ER chaperones and ATF4 [42]. Similarly, BiP/GRP78 was significantly enhanced in GBM cell lines, GBM patient samples and U87-derived mouse xenografts [43,44]. In addition, the expression level of BiP/GRP78 distinguished GBM (grade IV) from grade III glioma, and was related to the poor prognosis of GBM. Furthermore, metabolic flux analysis showed that ER stress enhances the uptake of glucose and glycolytic flux, thereby promoting the progress of GBM [44]. Moreover, compared with normal brain tissue, the activation of PERK branch was observed in grade III gliomas, especially in GBM samples. In vitro experiments have shown that PERK is essential for the survival of U87 and U251 cells [45]. These findings indicate that UPR activity in GBM is significantly elevated compared to normal tissue and contributes to the development of GBM. In our study, morusin could potentiate UPR and cause cytotoxicity in GBM cell lines and xenograft models. Human astrocytes were less sensitive to morusin, which supported the selective killing effect of morusin in GBM tumor cells. This provides a therapeutic window for the design of new UPR targeting strategies and the development of novel treatments for GBM.

UPR is composed of three different parallel signaling pathways or branches. These three branches are triggered by three different sensory ER transmembrane proteins: protein kinase R (PKR)-like ER kinase (PERK), inositol-requiring enzyme 1 (IRE1) and activating transcription factor 6 (ATF6). PERK dimerization results in autophosphorylation and subsequent phosphorylation of the eukaryotic translation initiator factor 2α (eIF2α), leading to inhibition of overall protein synthesis and thus preventing further accumulation of misfolded proteins in the ER. IRE1 comprises a serine/threonine-protein kinase domain and an endoribonuclease (RNase) domain and is activated upon dimerization and autophosphorylation. It catalyzes and produces functional X-box binding protein (XBP1). XBP1 is a transcription factor that regulates the activation of genes in ER-associated protein degradation (ERAD), protein folding and ER membrane expansion. ATF6 is a transmembrane transcription factor of the endoplasmic reticulum. ATF6 is activated by transport to the Golgi apparatus. In the Golgi apparatus, it is cleaved into two fragments by proteases to generate the cytoplasmic component ATF6f. ATF6f translocates to the nucleus, promoting the transcription of ERAD genes and XBP1 [46]. When cells are unable to cope with high level of ER stress, apoptosis is activated. UPR-dependent activation of the transcription factor C/EBPhomologous protein (CHOP) plays a vital role by modulating anti- and proapoptotic proteins, such as Bcl-2 family members [47]. We demonstrated that morusin significantly induced the expression of p-eIF2α, ATF6 and CHOP in a dose-dependent manner and led to GBM cell apoptosis. Further morphology analysis of ER in U87 and U251 cells confirmed the destruction of morusin to ER.

In addition to ERs and UPR, we also focused on the effect of morusin on GBM cell proliferation. Our results indicated that U87 and U251 cell cycles were arrested at G1/S phases after morusin treatment. The primary G1/S cell cycle checkpoint controls the commitment of eukaryotic cells to transition through the G1 phase to enter into the DNA synthesis S phase. As a critical G1/S checkpoint and a standard proliferation marker, cyclin D1 was dramatically inhibited by morusin in a dose-dependent manner. Another proliferation marker, PCNA, exhibited similar suppression after morusin treatment. Furthermore, we analyzed the phosphorylation level of the Akt–mTOR pathway because of its critical role in regulating diverse cellular functions, including metabolism, growth, proliferation, survival, transcription and protein synthesis. Morusin obviously inhibited the phosphorylation of Akt, mTOR and P70S6K, indicating that GBM cell growth, proliferation and many other cellular processes were suppressed by morusin.

GBM is considered as an incurable intracranial malignant tumor, having a median survival time of 15 months following an aggressive combination of therapies, including surgical resection and adjuvant radiation therapy (RT) with concurrent and adjuvant temozolomide (TMZ). However, it is well known that GBM is highly resistant to traditional chemotherapy and radiotherapy. Interestingly, it was found that the cytotoxic effects of traditional chemotherapy depend at least in part on the ER stress and UPR responses. For example, temozolomide (TMZ) appeared to induce BiP/GRP78 and CHOP, and knockdown of BiP/GRP78 in GBM cell lines in vitro enhanced CHOP activation and sensitized them to TMZ [43]. Likewise, radiotherapy-induced cell death in GBM cells was shown to be partly mediated by ER stress involving the PERK and IRE1 branches [48]. In addition, enhancing ER stress by hypoxia and celecoxib, a COX-2 inhibitor also known to activate ER stress, enhanced sensitivity for radiotherapy. Given that the cytotoxic effects of TMZ were found to depend at least in part on ER stress and UPR, ER stress and UPR targeting drugs may be effective in enhancing sensitivity for GBM chemotherapy and radiotherapy. In our research, we revealed that combinational treatment of morusin and TMZ dramatically induced ER stress and UPR, and thus inhibited cell viability of GBM cell lines. Furthermore, morusin has been shown to enhance the sensitivity of tumor to TMZ therapy in orthotopic GBM mouse models.

## 5. Conclusions

In summary, our results demonstrated that morusin could induce ER stress and UPR, arrest cell cycle at G1/S and inhibit cell proliferation through the AKT-mTOR pathway in GBM cells. Orthotopic xenograft models confirmed that morusin could elevate GBM tumor ER stress levels and suppress tumor progression in vivo. Moreover, morusin and TMZ synergistically induced ER stress and prolonged the overall survival time of tumor-bearing mice. Taken together, morusin may be a potential neoadjuvant chemotherapy or an alternative strategy for the treatment of GBM patients.

## Figures and Tables

**Figure 1 jcm-11-03662-f001:**
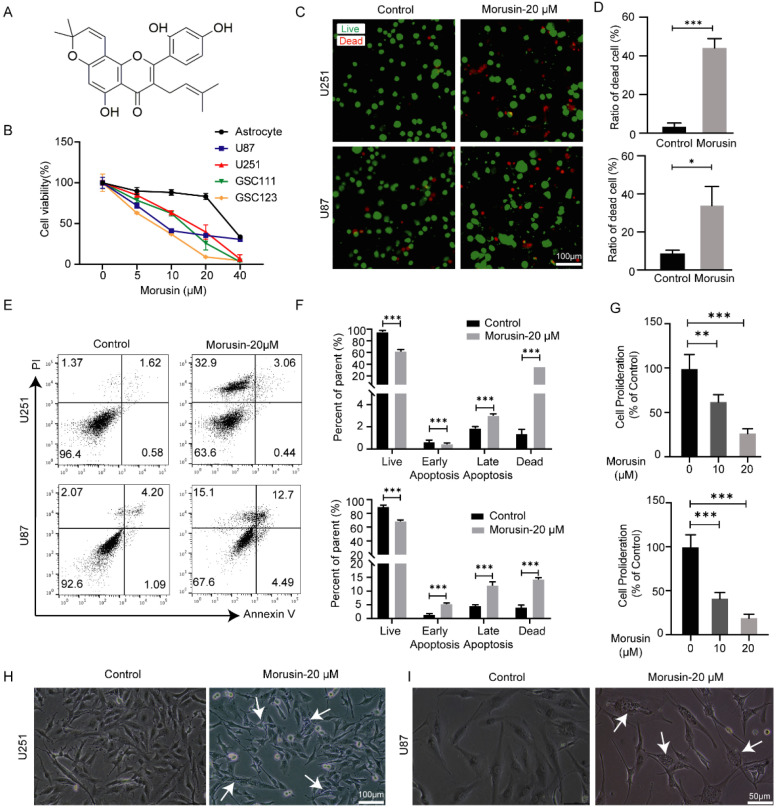
Morusin suppressed the cell viability and induced the apoptosis of GBM cells. (**A**) The structure of morusin. (**B**) Cell viabilities of GBM cells, GSC cells and normal human astrocytes after different concentrations of morusin exposure for 24 h. The results were calculated relative to the control group. (**C**) Representative images of live (green)/dead (red) fluorescent staining of U251 and U87 cells after morusin treatment (20 μM, 24 h). Bar = 100 μm (**D**) Quantification of cell death following live/dead fluorescent staining. (**E**) Annexin V/PI staining of morusin-treated GBM cells. Cells were exposed to morusin for 24 h. (**F**) Quantification of apoptotic cells following annexin V/PI staining. (**G**) Effects of morusin on U251 and U87 cell proliferation as assessed by cell counting assay. (**H**,**I**) Phase-contrast microscope showing morphological changes of U251 and U87 cells after treatment with 20 μM of morusin for 12 h. Bar = 100 μm. The data represent the mean ± SEM (*n* ≥ 3). * *p* < 0.05, ** *p* < 0.01, *** *p* < 0.0001 compared to indicated comparator.

**Figure 2 jcm-11-03662-f002:**
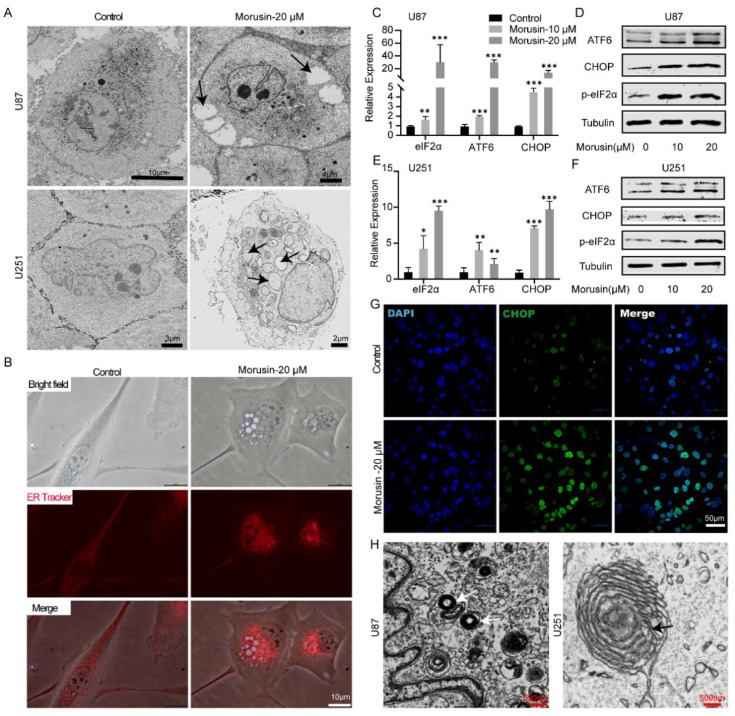
Morusin induced ER stress and UPR in GBM cells. (**A**) Cytoplasmic vacuolation in U87 and U251 cells after treatment with vehicle or 20 μM of morusin for 12 h. Cells were visualized via transmission electron microscopy. (**B**) Cytoplasmic vacuolation in GBM cells resulting from enlargement of the ER. U87 cells were incubated with ER-Tracker Red dye after exposure to morusin (20 μM) for 12 h and were observed via a fluorescence microscope. Bar = 10 μM. U87 and U251 cells were treated with morusin (0, 10, 20 μM) for 24 h and the transcriptional and protein levels of ER stress factors were determined by quantitative PCR and Western blot, respectively. Relative transcriptional expression levels of eIF2α, ATF6 and CHOP in U87 (**C**) and U251 (**E**) were normalized to GAPDH. Changes in protein expression of p-eIF2α, ATF6 and CHOP in in U87 (**D**) and U251 (**F**) following morusin were detected. (**G**) Immunofluorescent staining of CHOP (green) in U87 cells treated with morusin (20 μM) or vehicle for 12 h. Nuclei were stained with DAPI (blue). Bar = 50 μm. (**H**) Autophagosomes after morusin treatment (20 μM, 12 h) in U87 and U251 cells were visualized via transmission electron microscopy. Bar = 500 nm. The figures are representative data from at least three independent experiments. * *p* < 0.05, ** *p* < 0.01, *** *p* < 0.0001 compared to indicated comparator.

**Figure 3 jcm-11-03662-f003:**
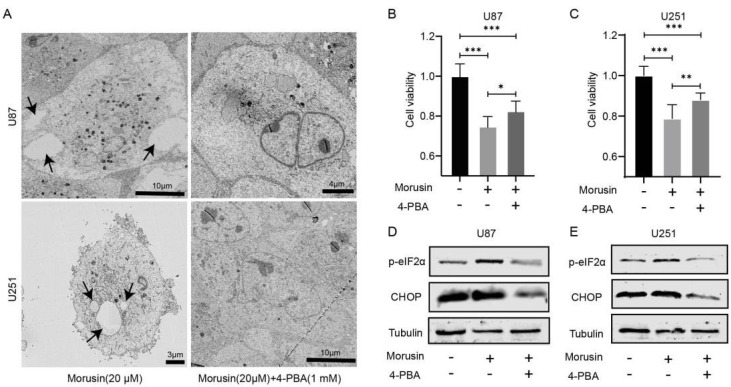
4-PBA rescued the cytotoxicity of morusin in GBM cells. (**A**) Cytoplasmic vacuolation in U87 and U251 cells after morusin (20 μM) and 4-BPA (1 mM) treatment for 12 h. Cells were visualized via transmission electron microscopy. (**B**,**C**) Cell viabilities of U87 and U251 cells after exposure to morusin (20 μM) with or without 4-BPA (1 mM) for 24 h. (**D**,**E**) Protein changes of p-eIF2α and CHOP in in U87 (D) and U251 (E) were determined by Western blot. Tubulin was used as a loading control. *n* = 3, * *p* < 0.05, ** *p* < 0.01, *** *p* < 0.0001 compared to indicated comparator.

**Figure 4 jcm-11-03662-f004:**
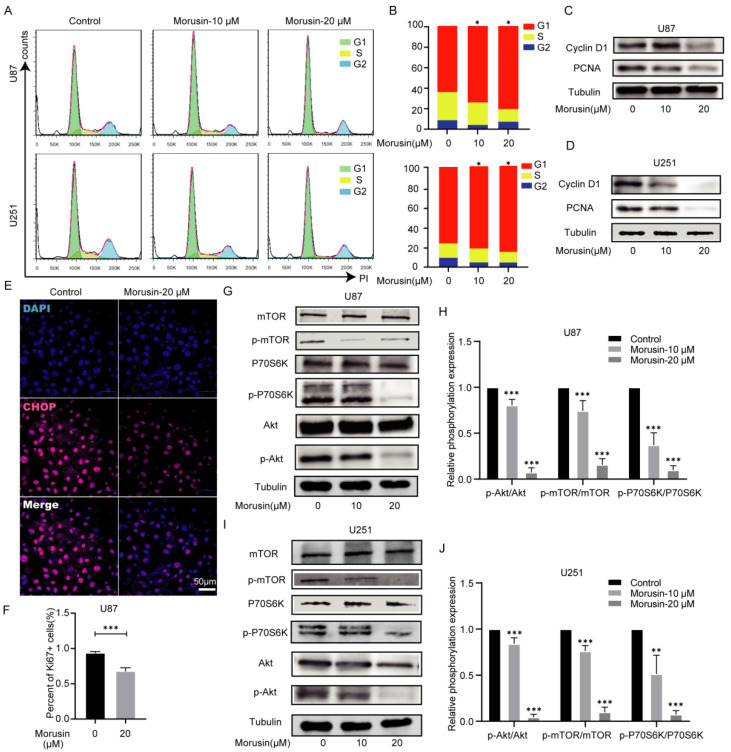
Morusin arrested cell cycle at G1 phase and inhibited cell proliferation through the Akt–mTOR–p70S6K pathway. (**A**,**B**) Cell-cycle profile of PI-stained cells was analyzed by flow cytometry (morusin treatment for 24 h after cell cycle synchronization). Quantification of cell cycle following PI staining. *n* = 3. (**C**,**D**) Protein expression levels of cyclin D1 and PCNA in U87 and U251 after 24 h morusin treatment (0, 10, 20 μM) were measured by Western blot. (**E**) Ki67 (red) was stained by Alexa Fluor 647 in U87 cells treated with morusin (20 μM, 12 h) or vehicle and analyzed by a confocal microscopy. Nuclei were stained with DAPI (blue). bar = 50 μm. (**F**) Quantification of the number of ki67 positive cells. (**G**–**J**) Down-regulation of Akt–mTOR–P70S6K pathway after 24 h morusin treatment (0, 10, 20 μM). Phosphorylation levels of AKT, mTOR and P70S6K were compared to total Akt, mTOR and P70S6K, respectively. * *p* < 0.05, ** *p* < 0.01, *** *p* < 0.0001 compared to indicated comparator.

**Figure 5 jcm-11-03662-f005:**
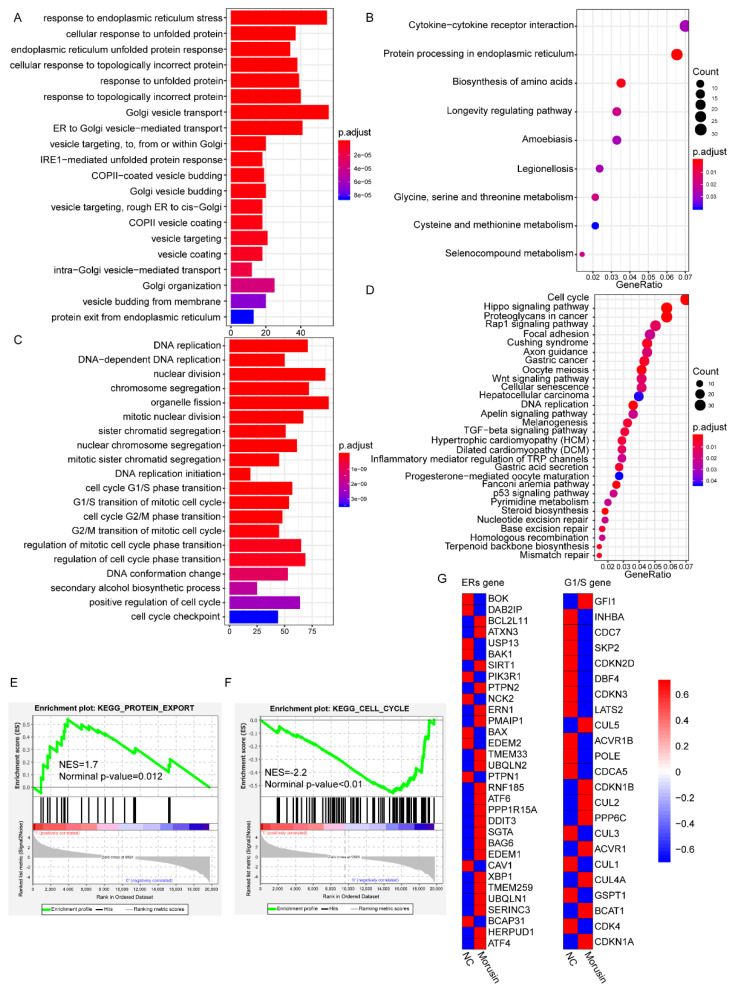
RNA-seq analysis of morusin treated U87 cells. (**A**,**B**) Gene Ontology (GO) and Kyoto Encyclopedia of Genes and Genomes (KEGG) pathway enrichment analysis of up-regulated mRNA after morusin treatment. (**C**,**D**) GO and KEGG pathway enrichment analysis of down-regulated mRNA. (**E**,**F**) GSEA-based KEGG enrichment plots of representative gene sets from activated pathway (protein export) and suppressed pathway (cell cycle). (**G**) Heatmap of mRNA expression levels in two KEGG gene sets (ERs and G1/S).

**Figure 6 jcm-11-03662-f006:**
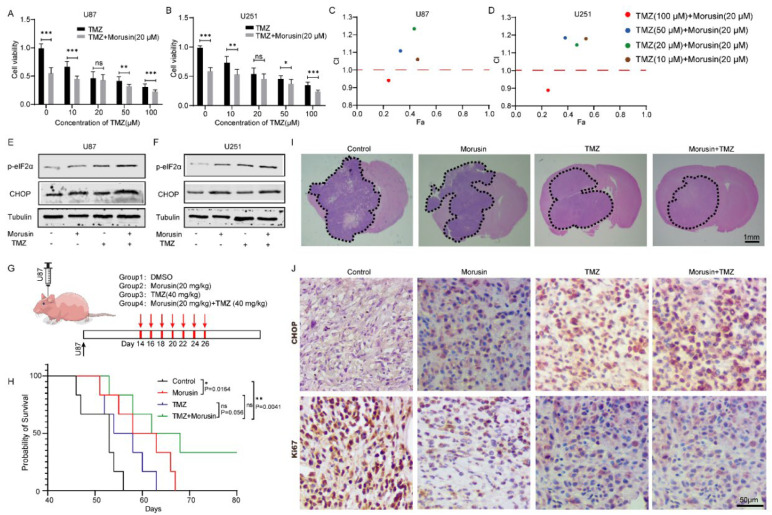
The synergistic anti-tumor effect of morusin and TMZ in GBM. (**A**,**B**) The viability of U87 and U251 cells treated with TMZ (0, 10, 20, 50, 100 μM) combined with morusin (20 μM) for 24 h. (**C**,**D**) Synergy was analyzed by CompuSyn software. CI = 1, >1 and <1 indicate additive, antagonistic and synergistic effects, respectively. (**E**,**F**) Levels of p-eIF2a and CHOP in GBM cells after morusin and TMZ exposure for 24 h. Tubulin was used as a loading control. *n* = 3. (**G**) A schematic of xenograft tumor experiment for the investigation of anti-tumor effects of morusin and TMZ in vivo. (**H**) The Kaplan–Meier survival curves of tumor bearing mice after treatment. (**I**) H&E-stained images of mouse brain sections. Scale bar = 1 mm. (**J**) Representative images of IHC analysis of CHOP and Ki-67 in the intracranial tumors after morusin and TMZ treatment. Immunoreactivity was detected by DAB chromogen (brown). The figures are representative data from at least three independent experiments. Scale bar = 50 µm. * *p* < 0.05, ** *p* < 0.01, *** *p* < 0.0001 compared to indicated comparator.

**Figure 7 jcm-11-03662-f007:**
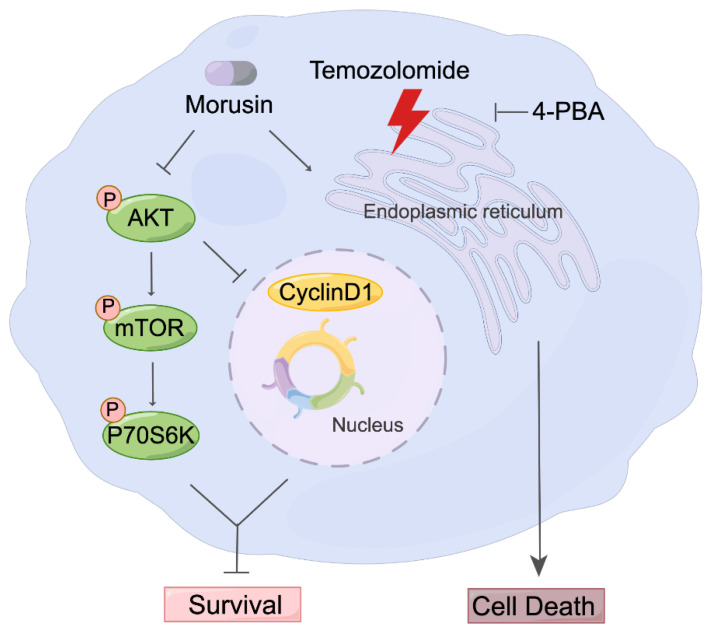
Schematic depicting the effect of morusin on GBM cells and the underlying mechanism.

**Table 1 jcm-11-03662-t001:** Primers used for qPCR.

Gene	Primer
ATF6	F:AGCAGCACCCAAGACTCAAAC
	R:GCATAAGCGTTGGTACTGTCTGA
eIF2α	F:TCGACCTCCTGAAGGCAGTT
	R:AGTTGTAGGTTGGGTATCCCAG
CHOP	F:GGAAACAGAGTGGTCATTCCC
	R:CTGCTTGAGCCGTTCATTCTC
GAPDH	F:AGAAGGCTGGGGCTCATTTG
	R:AGGGGCCATCCACAGTCTTC

## Data Availability

The datasets used and/or analyzed during the current study are available from the corresponding author on reasonable request.

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
