# Peer review of "Morusin Enhances Temozolomide Efficiency in GBM by Inducing Cytoplasmic Vacuolization and Endoplasmic Reticulum Stress"

_jcm, 2022, doi:10.3390/jcm11133662_

Round 1

Reviewer 1 Report

Zhao and co-workers presented a research article on the anti-tumor activity of the natural flavonoid compound Morusin in glioblastoma multiforme (GBM). They performed several in vitro and in vivo experimental procedures to demonstrate the cytotoxicity of morusin in GBM, and they demonstrated that morusin interfere with ER and arrest cell cycle at G1/s phase of the cell cycle. Moreover, they obtained results on the synergic effect of morusin in GBM cells when used in combination with temozolomide (TMZ) drug and they suggest that morusin could be used as a new chemotherapy compound for treatment of GBM patients.

The paper is well written, and the experimental results adequately support the conclusion of the authors. 

In addition, I have some minor comments for the authors:

Fig. 1I: the size of the U87 cells seem larger respect to U251 in panel H and U87/U251 in panel C. Probably it needs to correct the scale bar.

Fig. 2H: scale bars are not very well visible. I suggest adjusting their resolution and visibility.

Fig. 4: in line 274 is indicate a “cell cycle synchronization”: what protocol was used to synchronize cells? I suggest to add this in the Materials and Methods section.

Fig. 4E: images are at very low resolution, even if they were recorded by a CLSM. I suggest to show images with higher resolution (I suggest 300dpi).

Fig. 5E: characters are at very low resolution. I suggest redrawing panel E at higher resolution

Author Response

We appreciate these positive and critical comments from reviewer.

Reviewer 2 Report

Zhao et al. present convincing and interesting results regarding morusin treatment in glioblastoma. The manuscript is well constructed and clear.

Some aspects need to be addressed before possible publication.

Line 89: do you mean 104 cells? Please correct.

Line 103: Rrimers change to Primers

Specify if triplicates were used for in vitro experiments i.e. PCR, WB and RNA seq.

Line 126: define normal control (NC) cells meaning, untreated? Astrocytes?

Line 138: do you mean 106 cells? Please correct.

Line 138-139: specify if the cells where implanted right to the bregma.

Line 139-140: specify the concentration (mg/Kg) administered for Morusin and TMZ. Specify the % of DMSO used for injection. Why was the treatment started after intracranial implantation and not after around two weeks as it is usually done in GBM mouse models?

Author Response

We appreciate these positive and critical comments from reviewer
